# Gender Differences in Adverse Events Following the Pfizer-BioNTech COVID-19 Vaccine

**DOI:** 10.3390/vaccines10020233

**Published:** 2022-02-03

**Authors:** Manfred S Green, Victoria Peer, Avi Magid, Neta Hagani, Emilia Anis, Dorit Nitzan

**Affiliations:** 1School of Public Health, University of Haifa, Abba Khoushy 199, Haifa 3498838, Israel; victoriap@shamir.gov.il; 2Department of Health Systems Management, Max Stern Yezreel Valley College, Emek Yizreel 1930600, Israel; avim@yvc.ac.il; 3Rambam Medical Center, Haifa 3525408, Israel; nhag0171@uni.sydney.edu.au; 4Epidemiology Division, Israel Ministry of Health, Jerusalem 9446724, Israel; emilia.anis@moh.gov.il; 5World Health Organization, European Region, 2100 Copenhagen, Denmark; nitzand@who.int

**Keywords:** gender differences, COVID-19 vaccine, SARS-CoV-2, side-effects, adverse events

## Abstract

**Background:** The adverse events reported from the COVID-19 mRNA vaccines have varied from very mild, such as pain near the vaccination site, to more severe, with occasional anaphylaxis. Details of age-specific gender differences for the adverse effects are not well documented. **Methods:** Age and gender disaggregated data on reports of adverse events following two or three doses of the Pfizer-BioNTech COVID-19 vaccine were obtained from four cross-sectional studies. The first was from reports submitted to the Israel Ministry of Health national adverse events database (for ages 16 and above). The second was from a national cross-sectional survey based on an internet panel (for ages 30 and above), and the third and fourth were from cross-sectional surveys among employees of a large company (for ages 20–65) using links to a self-completed questionnaire. **Results:** In all studies, the risks of adverse events were higher following the second dose and consistently higher in females at all ages. The increased risk among females at all ages included local events such as pain at the injection site, systemic events such as fever, and sensory events such as paresthesia in the hands and face. For the combined adverse reactions, for the panel survey the female-to-male risk ratios (RRs) were 1.89 for the first vaccine dose and 1.82 for the second dose. In the cross-sectional workplace studies, the female-to-male RRs for the first, second and third doses exceeded 3.0 for adverse events, such as shivering, muscle pain, fatigue and headaches. **Conclusions:** The consistent excess in adverse events among females for the mRNA COVID-19 vaccine indicates the need to assess and report vaccine adverse events by gender. Gender differences in adverse events should be taken into account when determining dosing schedules.

## 1. Introduction

In general, reports on adverse events during vaccine trials are not disaggregated by gender [1,2]. Recent reports on anaphylaxis after receipt of mRNA vaccines are striking in the dominance of women. In a report from the national Vaccine Adverse Event Reporting System, there were 47 cases of anaphylaxis reported after receipt of the Pfizer-BioNTech COVID-19 vaccine, of whom 44 were women (94%). For the Moderna COVID-19 vaccine, 19 cases of anaphylaxis were reported, all of whom were women [3]. The majority of confirmed anaphylaxis cases (95%) were in females and occurred on the day of vaccination [4]. In a study of Mass General Brigham workers who received the first dose of an mRNA vaccine, 16 suffered from an episode of anaphylaxis, of whom 15 (94%) were women [5]. In another report, after the first dose of the Moderna COVID-19 vaccine (6), there were 10 anaphylactic reactions, all of whom were women. There were also 43 non-anaphylactic allergic reactions reported, of whom 39 (91%) were women [6].

Following the widespread introduction of mRNA vaccines, general gender differences in adverse events following vaccination have been reported, although this does not appear to have been systematically evaluated. For example, in the first month of COVID-19 mRNA vaccine safety monitoring in the USA, 79.1% of reports of adverse reactions were from women [7]. In contrast, myocarditis following mRNA COVID-19 vaccines has occurred primarily in young males [8,9]. For a number of vaccines, females tend to report more adverse reactions than males [10,11].

In the relatively few studies, the results were varied. In one study, there were no differences in the frequency and severity of adverse events associated with gender after the first and second doses of the vaccine [12]. In another study of an inactivated virus COVID-19 vaccine (Sinovac COVID-19), the frequency of adverse events was higher among female medical staff [13]. In a recent study on the 75 clinical trials on different types of COVID-19 vaccines, 24% reported their main results disaggregated by gender, and only 13% mentioned the implications of their study for females and males [14]. In a mini review of studies reporting safety outcomes of COVID-19 vaccines, the authors found a significant lack of gender-disaggregated evidence across studies [15].

Details of the gender differences in adverse events following COVID-19 vaccines by age group have not been well documented. Until very recently, the Pfizer-BioNtech COVID-19 vaccine was the only one in used in Israel. In this study, we examine and quantify gender differences in reports of adverse events occurring following the Pfizer-BioNtech COVID-19 vaccine using three separate sources of data from Israel.

## 2. Materials and Methods

### 2.1. Study Design

The data were obtained from three sources: passive reporting of adverse events following vaccination to the Israel Ministry of Health, a cross-sectional online panel survey conducted by a survey company and a cross-sectional study of employees of an industrial company. All vaccinees are requested to report all adverse events to the Ministry of Health, using a dedicated ministry internet site. Physicians can also report on that site. The details required are age, gender, date of vaccination, type of vaccine and type of adverse event. Data on the adverse events from the Pfizer-BioNTech vaccine in Israel in the period from December 2019 to June 2021 for ages 16 and above, disaggregated by gender and age, were obtained directly from the Israel Ministry of Health, Department of Epidemiology. During the period of the study, the Pfizer vaccine was the only one in use in Israel.

We also carried out three cross-sectional surveys during the vaccination campaign. The first was an online omnibus panel survey of 923 adults age 25 and older, carried out during June 2021, in which several questions on adverse events following the COVID-19 vaccine were included. The survey was carried out by a survey company that maintains a database of around 100,000 potential interviewees. The subjects included were those who had received the Pfizer-BioNtech vaccine and were age 30 and above. Details available were age, gender and the adverse events listed on a questionnaire. A total of 414 males and 509 females completed the survey.

The two other surveys were dedicated cross-sectional studies of 266 employees of a single workplace (152 males and 114 females, age 20–65), who had been vaccinated with the Pfizer-BioNTech vaccine in September 2021. Questionnaires were distributed online to the workers by the human resources department using an online link to the questionnaire available for a limited time. The questionnaire for the first survey contained detailed questions about possible adverse events following the first and second vaccine doses, and the second survey contained questions about possible adverse events following the third dose (172 males and 122 females, age 20–65), The questions were divided into categories as follows: local reactions (pain, swelling or redness at the injection site, pain over whole arm), systemic reactions (fever, shivering, muscle pains, joint pain, headache, fatigue, lack of ability to stand, general weakness, allergy and anaphylaxis), and sensory reactions (paresthesia in the injected hand or face, facial paralysis, herpes zoster).

The composition of the Pfizer-BioNTech COVID-19 vaccine reported to the FDA contains 30 mcg of a nucleoside-modified messenger RNA encoding the viral spike (S) glycoprotein of SARS-CoV-2. The vaccine also contains lipids including ((4-hydroxybutyl)azanediyl)bis(hexane-6,1-diyl)bis(2-hexyldecanoate),2-[(polyethylene glycol)-2000]-N,N-ditetradecylacetamide (ALC-0159), 1,2-distearoyl-sn-glycero-3-phosphocholine, and cholesterol, potassium chloride, monobasic potassium phosphate, sodium chloride, dibasic sodium phosphate dehydrate and sucrose. The vaccine does not contain preservatives.

### 2.2. Statistical Analyses

For the national data, graphical methods were used to describe the percentage of females and males reporting each adverse event for each vaccine dose and the ratio of female-to-male (F:M) percentages. For the internet panel and workplace cross-sectional studies, the percentage of vaccinees reporting each adverse event following each vaccine dose (also termed the risk) was calculated for each vaccine dose, by gender and age group, using the number of the adverse events reported divided by the number of vaccine doses. The age groups were divided into intervals as follows: 16–19, 20–29, 30–39, 40–49, 50–59, 60–69, 70–79, 80+. The ratio of the percentage of females to males reporting an adverse event was computed for each adverse event, with accompanying confidence interval and *p*-value. For the two cross-sectional studies, F:M ratios of the percentage of adverse events, together with confidence intervals and *p*-values, were computed using multinomial logistic regression (SAS version 9.4) and (SPSS version). Since there may be an impact from multiple testing, the *p*-values should be interpreted with caution.

## 3. Results

The summary of the adverse events reported to the Israel Ministry of Health following the first and second vaccine doses, for males and females separately, together with the F:M risk ratios (RRs) for each age group, are presented in Figure 1 and Figure 2. Data are shown for all adverse events and separately for local, neurologic and allergic events. Results for the first dose are shown in Figure 1.

Following the first dose (Figure 1), the risk of any event in all categories was higher in females at all age groups. The F:M RRs for any adverse event increased for allergic advents, from 1.4 in the youngest group to 3.72 at age 50–59 and then declining to 1.81 at age 80–89. The highest F:M RRs were for allergic events, ranging from 1.83 in the youngest to 4.77 at age 60–69. The age-related differences were not consistent for the allergic, local and neurological events, although the highest F:M RRs tended to occur between ages 40–69. Results for the second dose are shown in Figure 2.

The F:M RRs were higher following the second dose. For example, for any adverse event, the highest F:M RR was 4.79 in the age group 50–59. For allergic events, the F:M RR reached 14.2 at age 40–49 and 19.9 at age 50–59. For all categories, there was a strong age effect, with the lowest F:M RRs in at the extremes of the age groups.

After comparing the local and systemic side effects following receipt of three vaccines, it appears that the F:M ratios of systemic adverse events after the third vaccine are lower compared to the first and second vaccines (except for fever and joint pain, which were not observed after the first dose, and muscle pain, which was lower after the second dose).

Results of the online panel survey are shown in Table 1.

Overall, the F:M RRs for any moderate or severe adverse events were 1.89 and 1.82 for the first and second doses, respectively. The RRs were fairly consistent over the age groups. Overall, 5.7% of those who reported adverse events after the first dose sought medical care and 9.0% following the second dose. There were no obvious gender differences. Following the first dose, there were higher percentages in the age group 30–59, ranging between 5.6% and 9.5% compared to 3.2% at age 20–29 and 1.8% at age 60 and over.

### Adverse Events in the Workplace Survey

The adverse events in the workplace study were divided into three categories—local, systemic and sensory (neurological). For the local events, pain in the injected hand and rash in the injected area were significantly greater in females. In general, the reporting of almost all adverse events was higher in women than men. For specific side effects, the percentage of almost all adverse events was approximately double in females. The systemic events were all significantly greater in frequency in females, in particular for general weakness (F:M percent ratio of 3.43). Among the sensory events, the frequency of paresthesia in the injected hand and facial paralysis was greater in females. The results were generally similar after both the first and second doses.

Detailed results on the F:M ratios for moderate or severe adverse events were considered (as opposed to mild or none).

Almost all adverse events classified as moderate or severe were more frequent among females, following both the first dose and the second. For example, for pain all over the injected hand, the F:M RRs were 7.03 and 4.13 following the first and second doses, respectively. For shivering, the F:M RRs were 8.77 and 3.87 for the first and second doses, respectively. Other areas where the F:M RRs were particularly high were headache (9.15 and 3.28), fatigue (3.32 and 2.27) and difficulty in standing, which was reported almost exclusively in females (described in Table 2).

The summary (female-to-male ratio of the percentage) of the local and systemic adverse events reported following the first, second and third vaccine doses is presented in Figure 3. 

The frequency of adverse events lasting more than 24 h is shown in Table 3.

In general, the percentage reporting adverse events lasting more than 24 h was higher in females. This was observed particularly for pain in the affected hand, with an F:M RR of 3.98 and 3.58 for the first and second doses, respectively, shivering (5.25 and 4.26), headache (7.90 and 4.01) and fatigue (3.12 and 2.15). Difficulty in standing and general weakness were almost exclusively reported by females. Detailed results on the F:M ratios for moderate or severe adverse events after the third vaccine are described in Table 4.

The majority of adverse events following the third dose classified as moderate or severe were more frequent among females. For example, for pain all over the injected hand, the F:M RR was 4.23. For rash and redness in the injection site, the F:M RR was 3.52. The F:M RR for muscle pains was particularly high in females (3.25).

## 4. Discussion

Based on four studies in Israel, we found that adverse events following the Pfizer-BioNTech COVID-19 vaccine were substantially higher in females than in males. This was true for a wide range of adverse events, including local, systemic and sensory events. The results were consistent over three separate sources of data based on different methodologies. In addition, the severity and duration of the adverse events were greater in females.

There are potential biases in each of the four studies. The national reporting database is based on national data and a large number of vaccinees. However, the reporting is passive, and this could be a source of selection bias. Thus, we cannot exclude the possibility that women may be more likely than men to report adverse events that they experienced. In addition, it is clear that there is considerable under-reporting, and it is likely that only the more serious side effects are reported. This type of bias is much less likely to occur in the other studies, which were based on individual questionnaires. Information bias could occur if those with more severe symptoms are more likely to report them, and this could differ between males and females. Thus, this bias could be differential. However, in the cross-sectional workplace study, we evaluated gender differences in the severity of the adverse events. In the cross-sectional studies there may also be selection bias in those that choose to participate. Since the panel survey was not specifically aimed at collecting data on vaccine adverse events, there was no reason to believe that participation was affected by the presence or absence of adverse events. Another source of information bias is recall bias, which could affect the reporting of the milder adverse events. Despite the potential biases, the findings were consistent in the three sub-studies, one of which was based on passive reporting and the other two on active completion of questionnaires. This suggests that the gender differences found were real and not simply due to the possibility that women were more likely to report side effects in studies based on passive reporting.

The reports of the safety profile of the COVID-19 vaccine in the clinical trials generally have not provided analyses by gender; the adverse events are simply compared between the vaccine group and the placebo group. For example, in the report on the Pfizer vaccine, the safety profile of the vaccine was described as “short-term, mild-to-moderate pain at the injection site, fatigue, and headache, the incidence of serious adverse events as low and similar in the vaccine and placebo groups, and safety over a median of 2 months was similar to that of other viral vaccines” [1]. For the Moderna mRNA-1273 COVID-19 vaccine, the authors report that “moderate, transient reactogenicity after vaccination occurred more frequently in the mRNA-1273 vaccine group. Serious adverse events were rare, and the incidence was similar in the two groups” [2].

Following widespread vaccination, gender differences in adverse events following COVID-19 vaccines were reported (7). In the first month of COVID-19 vaccine safety monitoring in the USA, 79.1% of reports of adverse reactions were from women [7]. As mentioned, more than 90% of the reports of anaphylaxis following the mRNA vaccines have occurred in women [3]. Adverse events have also been more common in females in other type of vaccines against COVID-19. In a report on 30 cases of thromboembolic events following the AstraZeneca ChAdOx1-S– COVID-19 Vaccine, reported in the EEA, 19 (63%) were women [16]. Thrombotic events were reported in 10 cases and included deep vein thrombosis, hepatic vein thrombosis, mesenteric vein thrombosis, portal vein thrombosis and carotid artery thrombosis [16]. As previously mentioned, a notable exception is myocarditis following vaccination with the mRNA vaccines, which occurs predominantly in young males at rates of about one in six to ten thousand [9]. In our studies, the samples were not large enough to detect rare adverse events such as myocarditis.

The risk of adverse events following vaccines in general are consistently reported to be higher in females [17]. In the surveillance system for adverse effects following immunization (AEFI) in Victoria, Australia [18], females accounted for 55% of reports overall, and 80% of adults. Using vaccine safety claim databases, the risk of the 32 outcomes were often highest in females and adults ≥ 65 [19]. In a study of the risk of fever and rash following MMR vaccination in infants, it was higher in females, after controlling for background morbidity [20]. In a study of gender-specific differences in the adverse events following immunization with different vaccines in the Adverse Events Following Immunization (AEFI) reporting system in Ontario, Canada, between 2012 and 2015, the F:M reporting rate ratio (RRR) was 1.9 [21]. Gender differences were greatest in adults 18–64 years, with a RRR of 6.3. There were no gender differences in children <10 years. Adverse event F:M RRRs were highest for vaccines recommended for routine use in adults or high-risk populations. The highest event-specific F:M RRRs were for oculo-respiratory syndrome, anesthesia/paraesthesia and anaphylaxis. Serious adverse events were more commonly reported by females. In the computerized reporting registry of adverse events in Spain, there were more reports in females [22]. In a systematic review of seasonal influenza vaccine data, there were higher rates of adverse events following immunization in females [23].

The findings in our three studies using different methodologies should reduce the potential bias resulting from selective reporting by gender. Thus, there appears to be good evidence that the increased risk of adverse events in women following immunization with the Pfizer-BioNTech COVID-19 vaccine is real and not simply a reporting artifact. The mechanisms underlying the higher risk of adverse events in females are not fully understood. The lower COVID-19 case-fatality rates in women compared with men [24] may reflect an enhanced immune response in women. This could be part of the explanation of the higher frequency of adverse events in females due a stronger immediate response to the antigen, modulated through the innate immune system. Generally, females show higher expression of type IFN I, innate immune responses and T cell-associated genes [25]. Gender-related differences in immunity are influenced by X chromosome-linked genes and ChrY gene polymorphisms that are regulated by epigenetic mechanisms. In addition, hormonal factors could be important. The levels of sex hormones vary at different periods of life and affect immune cells [26]. Testosterone has the effect of depressing the innate and adaptive immune response [27,28]. Thus, it is conceivable that sex hormones are implicated in the mechanism of the increased reactogenicity of the vaccines in females [29]. Genetic factors could also play a part in the reactogenicity of the vaccines through an interaction with sex hormones [30]. The ACE2 and Ang-II receptor type 2 gene are both located on the X chromosome, and this may increase the immune response in females and increase the risks of vaccine-associated adverse events [31].

At the cellular level, a possible mechanism for the increased reactogenicity of vaccines in females has been proposed by Mackey et al. [30]. They reported that females are at increased risk of mast cell (MC)-associated diseases, including anaphylaxis [32]. They suggested that this could explain why males exhibit a significantly reduced severity of MC-mediated anaphylactic responses. They further suggested that that perinatal androgen exposure guides bone marrow MC progenitors toward a masculinized tissue MC phenotype, characterized by decreased concentration of prestored MC granule mediators such as histamine and a reduced mediator release upon degranulation. This would mean that gender differences in the MC phenotype are established in infancy.

## 5. Conclusions

In conclusion, the remarkably consistent excess in the rates of adverse events in females following immunization with the Pfizer-BioNTech COVID-19 vaccine, in all age groups, suggests that gender-specific factors influence the response to the vaccine. These findings indicate that different doses of the vaccine for men and women should be explored.

## Figures and Tables

**Figure 1 vaccines-10-00233-f001:**
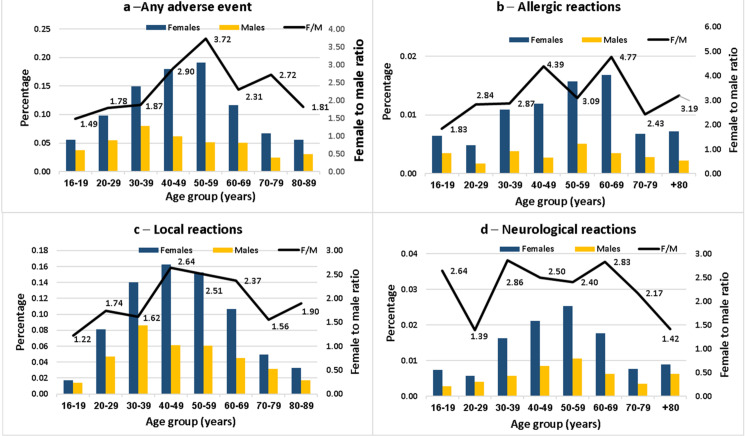
Female-to-male ratio of the percentage reporting adverse events after the first dose of the Pfizer-BioNTech COVID-19 vaccine in the Israel Ministry of Health database, by age group.

**Figure 2 vaccines-10-00233-f002:**
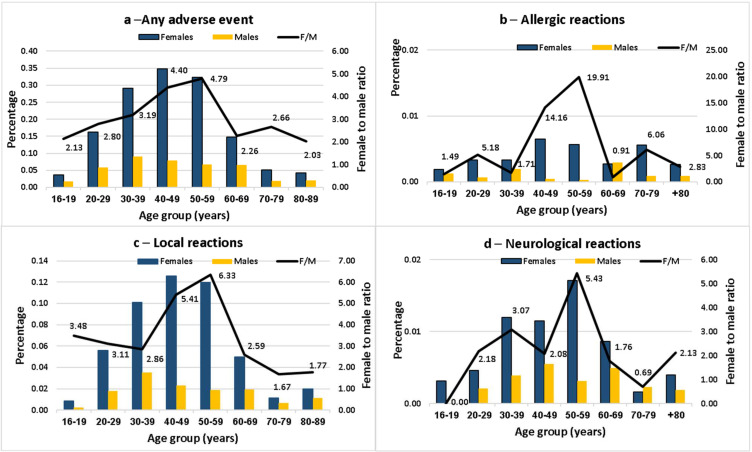
Female-to-male ratio of the percentage reporting adverse events after the second dose of the Pfizer-BioNTech COVID-19 vaccine in the Israel Ministry of Health database, by age group.

**Figure 3 vaccines-10-00233-f003:**
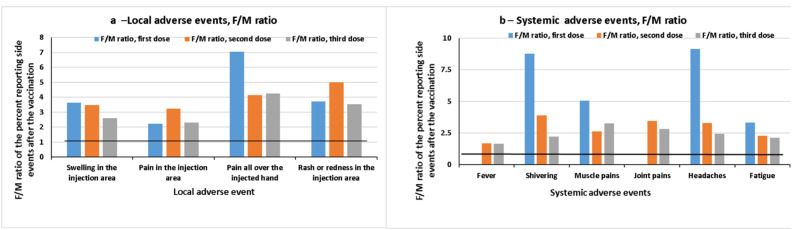
Female-to-male ratio of the percentage reporting local and systemic adverse events after three doses of vaccine.

**Table 1 vaccines-10-00233-t001:** Female-to-male ratio of the percentage reporting adverse events following each of two doses of the Pfizer-BioNTech COVID-19 vaccine and whether or not they sought medical care, in an online panel survey in Israel, by age group.

	First Dose							Second Dose						
	Male (M) (*N* = 414)		Female (F) (*N* = 509)		F:M Ratio	95% CI	*p*-Value	Male (M) (*N* = 414)		Female (F) (*N* = 509)		F:M Ratio	95% CI	*p*-Value
**Age**	** *n* **	**%**	** *n* **	**%**				** *n* **	**%**	** *n* **	**%**			
**20–29**	2	5.4	18	29.0	5.37	1.32–21.85	0.0046	13	35.1	28	45.2	1.29	0.77–2.16	0.3272
**30–39**	18	16.2	40	27.0	1.67	1.01–2.75	0.0498	35	31.5	69	46.6	1.48	1.07–2.04	0.0142
**40–49**	10	12.7	21	20.8	1.64	0.82–3.29	0.1515	14	17.7	39	38.6	2.18	1.28–3.72	0.0023
**50–59**	6	8.8	13	14.6	1.66	0.66–4.13	0.2709	9	13.2	23	25.8	1.95	0.97–3.94	0.0520
**60+**	10	8.4	15	13.8	1.64	0.77–3.49	0.1959	11	9.2	24	22.0	2.38	1.23–4.63	0.0075
**Total**	**46**	**11.1**	**107**	**21.0**	**1.89**	**1.37–2.61**	**<0.0001**	**82**	**19.8**	**183**	**36.0**	**1.82**	**1.45–2.28**	**<0.0001**
**Sought medical care**														
**Age**	** *n* **	**%**	** *n* **	**%**	**F** **:M**	**95% CI**	***p*-value**	** *n* **	**%**	** *n* **	**%**	**F** **:M**	**95% CI**	***p*-value**
**20–29**	5	13.5	2	3.2	0.24	0.05–1.17	0.0988	9	24.3	8	12.9	0.53	0.22–1.26	0.1499
**30–39**	3	2.7	14	9.5	3.50	1.03–11.88	0.0299	12	10.8	18	12.2	1.13	0.57–2.24	0.7366
**40–49**	7	8.9	6	5.9	0.67	0.24–1.92	0.4526	5	6.3	4	4.0	0.63	0.17–2.25	0.5086
**50–59**	2	2.9	5	5.6	1.91	0.38–9.55	0.6996	2	2.9	9	10.1	3.44	0.77–15.40	0.1156
**60+**	6	5.0	2	1.8	0.36	0.08–1.77	0.2842	5	4.2	7	6.4	1.53	0.50–4.67	0.4533
**Total**	**23**	**5.6**	**29**	**5.7**	**1.03**	**0.60–1.75**	**0.9259**	**33**	**8.0**	**46**	**9.0**	**1.13**	**0.74–1.74**	**0.5647**

**Table 2 vaccines-10-00233-t002:** Female-to-male ratio of the percentage reporting moderate or severe adverse events following the Pfizer-BioNTech COVID-19 vaccine in a cross-sectional study.

Adverse Events	First Dose							Second Dose						
	Male (*N* = 152)		Female (*N* = 114)		F to M Ratio	95% CI	*p*-Value	Male (*N* = 152)		Female (*N* = 114)		F to M Ratio	95% CI	*p*-Value
	** *n* **	**%**	** *n* **	**%**				** *n* **	**%**	** *n* **	**%**			
**Local**														
Swelling in the injection area	8	5.3	22	19.3	3.64	1.58–8.54	0.0001	8	5.3	21	18.4	3.47	1.42–6.80	0.0001
Pain in the injection area	41	27.0	69	60.0	2.22	1.42–3.54	0.0001	26	17.1	63	55.3	3.23	1.62–3.66	0.0001
Pain all over the injected hand	6	3.0	24	21.1	7.03	1.93–10.88	0.0001	8	5.3	25	21.9	4.13	1.68–7.72	0.0001
Rash or redness in the injection area	1	0.7	3	2.6	3.71	0.41–37.23	0.212	1	0.7	4	3.5	5.00	0.59–45.79	0.109
**Systemic**														
Fever	0	0	3	2.6	NA	NA	0.078	15	9.9	19	16.7	1.69	0.84–3.01	0.073
Shivering	2	1.3	13	11.4	8.77	1.81–34.28	0.0001	13	8.6	38	33.3	3.87	1.76–5.72	0.0001
Muscle pains	5	3.3	19	16.7	5.06	1.72–11.79	0.0001	21	13.8	41	36.0	2.61	1.35–3.51	0.0001
Joint pains	0	0	6	5.3	NA	NA	0.006	7	4.6	18	15.8	3.43	1.34–7.19	0.002
Headaches	3	2.0	21	18.3	9.15	2.45–26.35	0.0001	15	9.9	37	32.5	3.28	1.56–4.77	0.0001
Fatigue	10	6.6	25	21.9	3.32	1.45–5.85	0.0001	30	19.7	51	44.7	2.27	1.26–2.79	0.0001
Lack of ability to stand	0	0	10	8.8	NA	NA	0.0001	1	0.7	19	16.7	23.86	2.97–161.09	0.0001
General weakness	1	0.7	23	20.2	30.67	3.52–187.68	0.0001	19	12.5	37	32.5	2.60	1.33–3.67	0.0001
Allergy	0	0	3	2.6	NA	NA	0.078	0	0	1 0.9)	0.9	NA	NA	0.429
Anaphylaxis	0	0	0	0	NA	NA	NA	0	0	0	0	NA	NA	NA
**Sensory**														
Paresthesia in the injected hand	0	0	6	5.3	NA	NA	0.006	0	0	6	5.3	NA	NA	0.006
Facial paralysis	0	0	1	0.9	NA	NA	0.3	0	0	1	0.9	NA	NA	0.429
Facial paresthesia	0	0	2	1.8	NA	NA	0.183	0	0	2	1.8	NA	NA	0.183
Herpes zoster	0		1	0.9	NA	NA	0.429	1	0.7	0	0	NA	NA	0.571

**Table 3 vaccines-10-00233-t003:** Female-to-male ratio of the percentage reporting adverse events following the Pfizer-BioNTech COVID-19 vaccine and lasting more than 24 h, in a cross-sectional study.

Adverse Event	First Dose					Second Dose				
	Male (*N* = 152), *n* (%)	Female (*N* = 114), *n* (%)	F to M Ratio	95% CI	*p*-Value	Male (*N* = 152), *n* (%)	Female (*N* = 114), *n* (%)	F to M Ratio	95% CI	*p*-Value
**Local**										
Swelling in the injection area	29 (19.1)	44 (38.6)	2.02	1.15–2.64	0.0001	24 (15.8)	33(28.9)	1.83	0.99–2.60	0.008
Pain in the injection area	74 (48.7)	80 (70.2)	1.44	0.98–1.62	0.0001	49 (32.2)	69(60.5)	1.88	1.14–2.10	0.0001
Pain all over the injected hand	8 (5.3)	24 (21.1)	3.98	1.62–7.49	0.0001	9 (5.9)	24(21.1)	3.58	1.50–6.47	0.0001
Rash or redness in the injection area	2 (1.3)	5 (4.4)	3.38	0.64–16.39	0.123	1 (0.7)	4(3.5)	5.00	0.59–45.79	0.109
**Systemic**										
Fever	0 (0)	1 (0.9)	NA	NA	0.429	10 (6.6)	16(14.0)	2.12	0.94–4.25	0.035
Shivering	3 (2.0)	12 (10.5)	5.25	1.42–17.06	0.003	11 (7.2)	35(30.7)	4.26	1.84–6.60	0.0001
Muscle pains	8 (5.3)	17 (14.9)	2.81	1.16–5.82	0.007	16 (10.5)	38(33.3)	3.17	1.53–4.51	0.0001
Joint pains	1 (0.7)	3 (2.6)	3.71	0.41–37.23	0.212	5 (3.3)	15(13.2)	4.00	1.36–9.78	0.003
Headaches	3 (2.0)	18 (15.8)	7.90	2.12–23.39	0.0001	11 (7.2)	33(28.9)	4.01	1.75–6.34	0.0001
Fatigue	9 (5.9)	21 (18.4)	3.12	1.32–5.87	0.0001	28 (18.4)	45(39.5)	2.15	1.19–2.77	0.0001
Lack of ability to stand	1 (0.7)	7 (6.1)	8.71	1.10–70.97	0.012	3 (2.0)	15(13.2)	6.60	1.78–20.30	0.0001
General weakness	2 (1.3)	21 (18.4)	14.15	2.86–50.15	0.0001	15 (9.9)	30(26.3)	2.66	1.30–4.14	0.0001
Allergy	0 (0)	2 (1.8)	NA	NA	0.183	1 (0.7)	1 (0.9)	1.29	0.08–21.05	0.674
Anaphylaxis	0	0	NA	NA	NA	0	0	NA	NA	NA
**Sensory**										
Paresthesia in the injected hand	0 (0)	8 (7.0)	NA	NA	0.001	0 (0)	6(5.5)	NA	NA	0.006
Facial paralysis	1 (0.7)	0 (0)	NA	NA	0.571	1 (0.7)	1 (0.9)	1.29	0.08–21.05	0.674
Facial paresthesia	0 (0)	2 (1.8)	NA	NA	0.183	1 (0.7)	1 (0.9)	1.29	0.08–21.05	0.674
Herpes zoster	0 (0)	1 (0.9)	NA	NA	0.429	1 (0.7)	0(0)	NA	NA	0.571

**Table 4 vaccines-10-00233-t004:** Female-to-male ratio of the percentage reporting moderate or severe adverse events following the Pfizer-BioNTech COVID-19 third vaccine dose.

Adverse Event	Third Dose						
	Male (*N* = 172)		Female (*N* = 122)		F to M Ratio	95% CI	*p*-Value
	*n*	%	*n*	%			
**Local**							
Swelling in the injection area	19	11.0	35	28.7	2.59	1.56-4.31	0.0001
Pain in the injection area	46	26.7	75	61.5	2.29	1.73–3.05	<0.0001
Pain all over the injected hand	13	7.6	39	32.0	4.23	2.36–7.58	0.0008
Rash or redness in the injection area	4	2.3	10	8.2	3.52	0.47–10.98	0.0845
**Systemic**							
Fever	30	17.4	35	28.7	1.64	1.07–2.52	<0.0001
Shivering	21	12.2	33	27.1	2.21	1.35–3.63	<0.0001
Muscle pains	23	13.4	53	43.4	3.25	2.11–4.99	<0.0001
Joint pains	17	9.9	34	27.9	2.82	1.65–4.81	0.0002
Headaches	23	13.4	40	32.8	2.45	1.55–3.87	<0.0001
Fatigue	48	27.9	72	59.0	2.11	1.59–2.80	<0.0001
Lack of ability to stand	19	11.1	24	19.7	1.78	1.02–3.10	0.0004
General weakness	50	29.1	59	48.4	1.66	1.23–2.23	<0.0001
Allergy	7	4.1	3	2.5	0.60	0.16–2.29	0.1413
Anaphylaxis	3	1.7	0	0.0	-	–	-
**Sensory**							
Paresthesia in the injected hand	8	4.7	8	6.6	1.41	0.54–3.65	0.4797
Facial paralysis	3	1.7	1	0.8	0.47	0.05–4.46	0.3840
Facial paresthesia	3	1.7	1	0.8	0.47	0.05–4.46	0.3840
Herpes zoster	3	1.7	2	1.6	0.94	0.16–5.54	0.2693

## Data Availability

All data are available from the original sources or from the authors.

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
