# Peer review of "Gender Differences in Adverse Events Following the Pfizer-BioNTech COVID-19 Vaccine"

_vaccines, 2022, doi:10.3390/vaccines10020233_

Round 1

Reviewer 1 Report

The manuscript by Green et al. is properly prepared, the methodology is well prepared (you can possibly add a scheme for the recruitment of cohorts and their characteristics), the results are presented clearly, and the conclusions are supported by the obtained results. The methodology of statistical research is quite simplified, but adequate to the scope of the analysis performed.

In the discussion, the authors discuss well the weaknesses of the analysis and the possible influence of external factors on the observations, which is a valuable explanation. In my opinion, the possible reasons for the observation of the severity of adverse events in women are a bit too poorly explained and discussed. You can go a little "deeper" into immunology. The more so that it is assumed that in women there is a stronger response (which is somewhat inconsistent with the more and more frequently published results, as in the topic of vaccination against SARS-CoV-2 no differences in the humoral (measured by the level of antibodies) or cellular responses (measured e.g. on the basis of INF gamma or cytometrically) are observed.

My next comment concerns the mentions in the introduction and the discussion of other vaccines. In my opinion, they should be removed so that the text is consistent. This manuscript relates only to the Pfizer/BioNtech vaccine and should be commented on in this regard.

The manuscript also requires careful editing. 

Author Response

We thank the reviewer for the positive comments. The following are our responses to the reviewer.

 Reviewer 1

Comment: You can go a little "deeper" into immunology. The more so that it is assumed that in women there is a stronger response (which is somewhat inconsistent with the more and more frequently published results, as in the topic of vaccination against SARS-CoV-2 no differences in the humoral (measured by the level of antibodies) or cellular responses (measured e.g. on the basis of INF gamma or cytometrically) are observed.

Response: As regards the question of the possible mechanisms, since our studies did not specifically address the mechanisms, we believe that we should not go too far in our explanations. However, we have made this clearer in the Discussion.

Comment: My next comment concerns the mentions in the introduction and the discussion of other vaccines. In my opinion, they should be removed so that the text is consistent. This manuscript relates only to the Pfizer/BioNtech vaccine and should be commented on in this regard.

Response: We believe it is important to put our findings for the Pfizer mRNA vaccine in the context of what has been reported for other types of vaccines. We have now made this clearer in the Introduction and Discussion.

 Comment: The manuscript also requires careful editing. 

Response: The manuscript has undergone editing.

Reviewer 2 Report

Data on adverse events were gleaned from various sites in Isreal where the Pfizer-BionTech vaccines have been rolled out with two or three vaccinations.

It was found that there is a marked difference between adverse events in women and men. The limitations of the data sources are well addressed in terms of reporting bias.

This is an important study, suggesting different doses for males and females to be researched.

Queries:

For consistency I would suggest the term "gender-based" is used throughout the paper and not "sex-based". Or vice versa.

There is a typo in line 150 that needs to be addressed.

Author Response

We thank the reviewer for the positive comments.

Comment: For consistency I would suggest the term "gender-based" is used throughout the paper and not "sex-based". Or vice versa.

Response: We accept your comment and used the term ''gender'' throughout the paper.

 Comment: There is a typo in line 150 that needs to be addressed.

Response: The manuscript has undergone editing.

Reviewer 3 Report

Authors report adverse events following the Pfizer BioNtech COVID-19 vaccine found that are higher in females than in males. I think that is a good study but with some biases reported by the same authors. Would have been interested to know the clinical status of subjects involved in this study, it could be a starting point for a subsequent study.  I believe that this paper is suitable for publication.

Author Response

Thank YOU for your  comments.